# Genome-Wide Analysis Reveals Changes in Long Noncoding RNAs in the Differentiation of Canine BMSCs into Insulin-Producing Cells

**DOI:** 10.3390/ijms21155549

**Published:** 2020-08-03

**Authors:** Jinglu Wang, Pengxiu Dai, Dengke Gao, Xia Zhang, Chenmei Ruan, Jiakai Li, Yijing Chen, Luwen Zhang, Yihua Zhang

**Affiliations:** The College of Veterinary Medicine of the Northwest Agriculture and Forestry University, No.3 Taicheng Road, Yangling 712100, China; 2016060212@nwsuaf.edu.cn (J.W.); dpx910405@163.com (P.D.); gdk960101@163.com (D.G.); 17792525977@163.com (X.Z.); rcm900213@163.com (C.R.); llijiakai@163.com (J.L.); 2018120345@163.com (Y.C.); 2020237584@163.com (L.Z.)

**Keywords:** cBMSC, lncRNA, sequencing, IPCs, transdifferentiation

## Abstract

Long noncoding RNAs (lncRNAs) have been extensively explored over the past decade, including mice and humans. However, their impact on the transdifferentiation of canine bone marrow mesenchymal stem cells (cBMSCs) into insulin-producing cells (IPCs) is largely unknown. In this study, we used a three-step induction procedure to induce cBMSCs into IPCs, and samples (two biological replicates each) were obtained after each step; the samples consisted of “BMSCs” (B), “stage 1” (S1), “stage 2” (S2), “stage 3” (S3), and “islets” (I). After sequencing, 15,091 lncRNAs were identified, and we screened 110, 41, 23, and 686 differentially expressed lncRNAs (*p*_adjusted_ < 0.05) in B vs. S1, S1 vs. S2, S2 vs. S3, and I vs. S3 pairwise comparisons, respectively. In lncRNA target prediction, there were 166,623 colocalized targets and 2,976,362 correlated targets. Gene Ontology (GO) analysis showed that binding represented the main molecular functions of both the cis- and trans-modes. Kyoto Encyclopedia of Genes and Genomes (KEGG) analysis suggested that the insulin signaling pathway, Rap1 signaling pathway, tight junctions, MAPK signaling pathway, and cell cycle were enriched for these relative genes. The expression of lncRNAs was verified using qRT-PCR. This study provides a lncRNA catalog for future research concerning the mechanism of the transdifferentiation of cBMSCs into IPCs.

## 1. Introduction

Long noncoding RNAs (LncRNAs) are the rest of the protein-coding transcripts >200 nt in length that lack coding potential [1]. LncRNAs were discovered many years ago; for example, XIST was found in the 1980s [2]. To date, lncRNAs can be divided into five kinds based on their relative positions with respect to protein-coding genes: (1) intergenic lncRNAs, which are also known as lincRNAs, are transcribed from the DNA sequence between two protein-coding genes [3,4]; (2) intronic lncRNAs are transcribed from introns of protein-coding genes [5]; (3) antisense lncRNAs, as their name implies, are generated from the opposite direction of protein-coding genes [6]; (4) sense lncRNAs overlap protein-coding genes; and (5) bidirectional lncRNAs [7]. LncRNA catalogs of human [8], mouse [9], zebrafish [10], and fly [11] lncRNAs have already been collected; however, canine lncRNAs data are lacking. LncRNAs can be classified into two categories based on whether they function in cis- (these lncRNAs are located near the transcription site) or in trans- (this role is more complicated than the cis-role). These potential differences provide lncRNAs the ability to participate in many biological processes. It has been reported that lncRNAs can be therapeutic targets for managing many diseases, such as breast cancer [12] and hepatocellular carcinoma [13]. LncRNAs can also influence the transcription of genes through antisense-mediated transcript knockdown, which eventually induces premature transcription termination [14]. Due to their posttranscriptional surveillance properties, lncRNAs play a key role in controlling various cellular functions [15]. Another study has indicated that lncRNAs can modulate multiple pathways, including the mTOR signaling pathway [16]. In addition, many lncRNAs are involved in the differentiation of various cells; for example, TINCR can control somatic tissue differentiation [17] and the osteogenic differentiation of mesenchymal stem cells [18,19].

The prevalence of type I diabetes is currently rapidly increasing and is associated with the lack of islet sources for treatment, rendering the attainment of islets from other cell types indispensable. Moreover, as of 2020, diabetes mellitus has consistently been among the most common comorbidities found in patients with COVID-19 [20]. As companion animals living alongside humans, canines constitute an ideal model for diabetes research, although such a model has long been overlooked because of the lack of data in this field. In our study, we focused on the lncRNA sequences of samples generated from different stages of the transdifferentiation of canine bone marrow mesenchymal stem cells (cBMSCs) into insulin-producing cells (IPCs). To the best of our knowledge, this article presents the first attempt to conduct a mechanistic study based on the lncRNA profiling of cBMSCs according to the literature.

## 2. Results

### 2.1. Overview of lncRNA Sequencing

The library was constructed from B, I, S1, S2, and S3, with two biological replicates. The sequencing data showed that there were approximately 90 million clean reads of approximately 100 million raw reads per sample (Appendix A). Among these reads, 92–95% of the clean reads mapped to the reference genome (*Canis lupus familiaris* CanFam3.1, NCBI annotation release number 105), and the percentage of uniquely mapped reads accounted for 78.95–88.65% (Appendix A). The mapped reads were classified into eight types, and the percentages of lncRNAs in B, I, S1, S2, and S3 were 0.4%, 0.9%, 0.5%, 0.7%, and 0.7%, respectively (Appendix A). The data were uploaded to the SRA (Sequence Read Archive) database (https://www.ncbi.nlm.nih.gov/sra) under submission number SUB7457848.

### 2.2. Identification of lncRNAs

We used a 5-step procedure to filter the correct lncRNAs without gene-coding transcripts (Figure 1) and screened 15,091 lncRNAs including lincRNAs (30.1%), antisense lncRNAs (8.5%), and intronic lncRNAs (61.4%) (Figure 2A). Most lncRNAs contained approximately 4 exons and no more than 20 exons; moreover, the peak number in the protein-coding transcripts was 8 exons, with a maximum of 40 exons (Figure 2C). In addition, a length of 1000 bp represented the dominant portion of lncRNAs, while the length of known transcripts is nearly 2500 bp (Figure 2D). Overall, compared with protein-coding transcripts, lncRNAs had fewer exons but a similar length.

### 2.3. Differentially Expressed lncRNAs

Pairwise comparisons between different samples based on the fragments per kilobase of exon per million fragments mapped (FPKM) indicated that all samples could be clustered into different groups (two biological replicate each) and showed that I and S3 were similar to some extent (Figure 2B). There were 21 upregulated and 89 downregulated lncRNAs in the B vs. S1 comparison (Figure 3A). Similarly, there were 20 upregulated and 21 downregulated lncRNAs in S1 vs. S2 (Figure 3B), 12 upregulated and 11 downregulated lncRNAs in S2 vs. S3 (Figure 3C), 357 upregulated and 329 downregulated lncRNAs in the I vs. S3 comparison (Figure 3D), and 198 upregulated and 1196 downregulated lncRNAs in the B vs. I comparison (Figure 3E). These findings suggest that the differentially expressed lncRNAs slightly declined during induction (Figure 3F).

### 2.4. Enrichment Analysis of Colocalized Genes

Gene Ontology (GO) analysis of the differentially expressed colocalized genes showed that, in the B vs. S1 comparison, purine nucleoside binding (GO:0001883), enzyme regulator activity (GO:0030234), and small-molecule binding (GO:0036094) constituted the majority (Figure 4A); moreover, many other types of binding activity were significant. However, the S1 vs. S2 comparison was enriched in genes involved in protein complexes (GO:0043234) and macromolecular complexes (GO:0032991) (Figure 4B). The I vs. S3 comparison was enriched in genes involved in protein folding (GO:0006457), cyclase regulator activity (GO:0010851), and guanylate cyclase regulator activity (GO:0030249) (Figure 4D), and the S2 vs. S3 comparison was marginal based on the corrected *p* value (Figure 4C). The results of a Kyoto Encyclopedia of Genes and Genomes (KEGG) analysis revealed the following: ECM–receptor interactions (cfa04512), the PI3K-Akt signaling pathway (cfa04151), and the MAPK signaling pathway (cfa04010) in the B vs. S1 comparison (Figure 5A and Appendix A); the PI3K-Akt signaling pathway (cfa04151), the AMPK signaling pathway (cfa04152), the insulin signaling pathway (cfa04910), and ribosomes (cfa03010) in the S1 vs. S2 comparison (Figure 5B and Appendix A); the cell cycle (cfa04110), platelet activation (cfa04611), and metabolic pathways (cfa01100) in the S2 vs. S3 comparison (Figure 5C and Appendix A); and the cell cycle (cfa04110), the T cell receptor signaling pathway (cfa04660), and type I diabetes mellitus (cfa04940) in the I vs. S3 comparison (Figure 5D and Appendix A).

### 2.5. Enrichment Analysis of Coexpressed Genes

The genes in which expression levels paralleled those of lncRNAs were considered co-expressed genes (Pearson’s correlation coefficient > 0.95). In the GO analysis of the B vs. S1 comparison, these genes were enriched in intracellular signal transduction (GO:0035556), enzyme regulator activity (GO:0030234), protein tyrosine phosphatase activity (GO:0004725), and binding (GO:0005488) (Figure 6A); in the S1 vs. S2 comparison, these genes were enriched in signal transduction (GO:0007165), GTPase regulator activity (GO:0030695), cell communication (GO:0007154), and regulation of cellular process (GO:0050794) (Figure 6B); in the S2 vs. S3 comparison, these genes were enriched in G-protein coupled receptor activity (GO:0004930), transmembrane signaling receptor activity (GO:0004888). and cell surface receptor signaling pathway (GO:0007166) (Figure 6C); and in the I vs. S3 comparison, these genes were enriched in binding (GO:0005488), intracellular signal transduction (GO:0035556), and small GTPase-mediated signal transduction (GO:0007264) (Figure 6D). KEGG analysis demonstrated that the B vs. S1 comparison was enriched in genes involved in maturity-onset diabetes of the young (cfa04950), pancreatic secretion (cfa04972), and the Rap1 signaling pathway (cfa04015) (Figure 7A), while theS1 vs. S2 comparison was enriched in maturity-onset diabetes of the young (cfa04950), cell adhesion molecules (CAMs) (cfa04514), and the Rap1 signaling pathway (cfa04015) (Figure 7B). The S2 vs. S3 comparison was not very significant (Figure 7C), and the I vs. S3 comparion was enriched in the PI3K-Akt signaling pathway (cfa04151) and the Rap1 signaling pathway (cfa04015) (Figure 7D).

### 2.6. Verification of Target Genes and lncRNAs

*FOS* (Fos proto-oncogene), *VIP* (vasoactive intestinal peptide), *SSTR2* (somatostatin receptor 2), and *RPS6KA6* (ribosomal protein S6 kinase A6) were selected from the PPI (protein–protein interaction) network based on our previous work of protein-coding transcripts, and the chosen lncRNAs (LNC_000123, LNC_004661, LNC_014532, LNC_004320, and XR_292983.1) were colocalized with these genes. The qRT-PCR results were consistent with the RNA-Seq data (Figure 8). However, whether internal interactions exist requires further detection.

## 3. Discussion

Nearly 90% of the mammalian genome is transcribed as noncoding RNAs, and many of these noncoding RNAs are treated as junk or transcript noise [21]. However, this concept has been brought to the forefront in recent years. It has been confirmed that lncRNAs can function in transcription and posttranscription in addition to interacting with proteins [22] and can even participate in epigenetic modification [23]. According to many studies in other species and based on existing high-throughput technology, many lncRNAs have been reported in either disease research or basic research areas. The two reasons we chose cBMSCs, derived from the long bones of Chinese rural dogs, as our cell resource were as follows: (1) cBMSCs, which have the ability to evade the surveillance of the immune system [24] and are used as materials in diabetes research, are not involved in ethical problems as embryonic stem cells or other somatic cells derived from humans. (2) Canines, which are our friends and family members in normal life, represent an ideal model for disease research because they live in the same environment as humans; furthermore, there are increasing numbers of clinical cases of canine diabetes. In general, our work can serve as foundation not only for canine clinical treatment but also for research investigating human diabetes.

In this study, based on a five-step filter (Figure 1), we obtained 15,091 lncRNAs and 110, 41, 23, and 686 differentially expressed lncRNAs in the pairwise comparisons of different states of induction. LncRNAs accounted for only less than one percent of the RNA-Seq data, whereas protein-coding transcripts represented nearly 70%. However, we observed a relatively large increase in the lncRNAs ratio during the induction (Appendix A). Other research has shown that a low number of lncRNAs is sensible [25,26]. Regarding the characteristics of lncRNAs, their exons and lengths are smaller and shorter than those of protein-coding transcripts [27]; as a result, lncRNAs cannot be translated into proteins, although some lncRNAs can encode functional peptides to regulate biological processes [28]. The expression levels were verified by qRT-PCR, and the trends were consistent with the RNA-Seq data; however, the expression level of the lncRNAs was much lower than that of the protein-coding transcripts (Figure 8). The genes we chose in this study were derived from differentially expressed genes that also had an interaction network with several key genes, including *INS* (insulin), *SST* (somatostatin), and *GCG* (glucagon). *FOS* and *RPS6KA6* are members of the MAPK signaling pathway, which is a vital pathway for cell differentiation and proliferation [29,30]. Tuning these genes may contribute to the transdifferentiation of cBMSCs into IPCs. Therefore, we chose five lncRNAs that were colocalized with these genes to verify their expression levels, and the results confirmed the profile data, although whether these lncRNAs have internal relationships needs to be explored in depth. GO analysis showed that. during induction binding, enzyme regulatory activity and signal transduction were the main activities. Because islets are constructed by endocrine cells, the Golgi complex remains highly dynamic and the enzymes involved vary [31,32]. KEGG analysis identified the MAPK signaling pathway, PI3K-Akt signaling pathway, insulin signaling pathway, cell cycle, and Rap1 signaling pathway, all of which are associated with pancreas development [33,34,35].

Our work demonstrates the characteristics, changes, and potential function of lncRNAs during the transdifferentiation of cBMSCs into IPCs. These findings expand the canine lncRNA database and could also serve as a reliable foundation for further research.

## 4. Materials and Methods

### 4.1. Separation, Cultivation, and Induction of cBMSCs

The cBMSCs and islets used in this experiment were conserved in our laboratory and identified by flow cytometry and a three-lineage differentiation procedure, and the separation procedure was described previously [36]. We also used a previously reported three-step induction method to convert cBMSCs into IPCs [37]. The cBMSCs were placed in a suspended state to form spheroids, which mimicked the characteristics of islets in the pancreas. The dishes used in the protocol were all treated with 2-hydroxyethylmethacrylate (Sigma-Aldrich, St. Louis, MO, USA). In S1, a passage of 4 cells were cultured in Dulbecco’s modified Eagle’s medium (DMEM) containing high glucose (17.5 mmol/L glucose; Hyclone, Logan, UT, USA), 10 ng/mL basic fibroblast growth factor (bFGF; Invitrogen, Carlsbad, CA, USA), 10 ng/mL epidermal growth factor (EGF; Chemicon, CA, USA), 2% B27 supplement minus insulin (Gibco, Invitrogen, Paisley, UK), 0.5% bovine serum albumin (BSA; Solarbio, Beijing, China), and 0.1 mmol/L β-mercaptoethanol. After two days, the medium was exchanged for the S2 induction medium and supplied with DMEM containing high glucose, 10 ng/mL EGF, 20 ng/mL Activin A (Peprotech, Offenbach, Germany), 10 mmol/L nicotinamide (Sigma, St. Louis, MO, USA), 2% B27, 0.5% BSA, and 0.1 mmol/L β-mercaptoethanol for 4 days. During S3, which also lasted for 4 days, the spheroids were suspended in DMEM containing low glucose (5.6 mmol/L glucose), 10 ng/mL EGF, 10 nmol/L exendin-4 (Peprotech), 10 ng/mL betacellulin (Peprotech), 2% B27, 0.5% BSA, and 0.1 mmol/L β-mercaptoethanol. The medium was changed every 2 days.

### 4.2. RNA Isolation and Qualification

According to the manufacturer’s protocol, total RNA was extracted using TRIzol^®^ reagent (Invitrogen, USA), and genomic DNA was removed using DNase I (Takara, Japan). RNA purity was checked using a NanoPhotometer^®^ spectrophotometer (IMPLEN, CA, USA), and the RNA concentration was measured using a Qubit^®^ RNA Assay Kit in conjunction with a Qubit^®^ 2.0 Fluorometer (Life Technologies, CA, USA). RNA integrity was assessed using an RNA Nano 6000 Assay Kit in conjunction with a Bioanalyzer 2100 system (Agilent Technologies, Santa Clara, CA, USA).

### 4.3. Library Preparation

For RNA sample preparations, the input material was 3 μg of RNA per sample. First, the ribosomal RNA was removed by an Epicentre Ribo-zero™ rRNA Removal Kit (Epicentre, Brooklyn, NY, USA), and ethanol precipitation was performed to obtain rRNA-free RNA. Subsequently, the libraries were constructed by using an NEBNext^®^ Ultra™ Directional RNA Library Prep Kit for Illumina^®^ (NEB, Ipswich, MA, USA) according to the manufacturer’s instruction. Then, we acquired RNA fragmentation by using NEBNext First Strand Synthesis Reaction Buffer (5×) under an increased temperature. The first-strand cDNA was subsequently synthesized using random hexamer primers; then, DNA polymerase I and RNase H were used for second-strand cDNA synthesis with dNTPs and dTTP substituted by dUTP in the reaction buffer. Simultaneously, exonuclease/polymerase activities turned all remaining overhangs into blunt ends. Then, the NEBNext adaptors with hairpin loop structures were linked to the 3′ ends of cDNA fragments that were previously acetylated. In addition, cDNA fragments 150–200 bp in length were screened by an AMPure XP system (Beckman Coulter, Beverly, MA, USA). Before PCR, 3 μL USER Enzyme (NEB, USA) was added to the cDNA at 37 °C for 15 min and at 95 °C for 5 min. PCR was then performed with Phusion High-Fidelity DNA polymerase, Universal PCR primers and Index (X) Primer. The PCR products were finally purified with an AMPure XP system and assessed by an Agilent Bioanalyzer 2100 system.

## 5. Sequencing

The TruSeq PE Cluster Kit v3-cBot-HS (Illumina) was used to cluster the index-encoded samples using a cBot cluster generation system. Following the cluster generation, sequencing was performed on an Illumina HiSeq 4000 platform, and 150 bp paired-end reads were generated.

### 5.1. Transcriptome Assembly

The mapped reads of each sample were assembled by StringTie (v1.3.1) [38] in accordance with the method in the reference. By using the high-quality assembler StringTie, we obtained potential transcripts with full-length and splice variants for each gene locus.

### 5.2. Identification of lncRNAs

We used CNCI (Coding-Non-Coding-Index) (v2) [39] with the default parameters. We also used the CPC (Coding Potential Calculator) (0.9-r2) [40] and the NCBI eukaryote protein database, with the e-value set to “1e-10” in our analysis. We also used Pfam Scan (v1.3) [41,42], with the following default parameters: -E 0.001 --domE 0.001. We performed multispecies genome sequence alignments and used PhyloCSF [43] with the default parameters. PHAST (v1.3) is a software package that contains many statistical programs and is mostly used for phylogenetic analyses [44], and phastCons is a conservation scoring and identification program for conserved elements. We used phyloFit to construct phylogenetic models of the conserved and nonconserved regions of species and then exported to phyloP with the model and hidden Markov model (HMM) transition parameters to perform the conservation analysis of the lncRNAs and protein coding genes.

### 5.3. Differential Expression Analysis

FPKM is calculated based on the length of the fragments and the read count mapped to these fragments. Cuffdiff (v2.1.1) was used to calculate the FPKMs of both the lncRNAs and coding genes in each sample. The FPKMs of the genes were computed by summing the FPKMs of the transcripts in each gene group. Transcripts with *p*_adjust_ < 0.05 were considered differentially expressed.

### 5.4. GO and KEGG Enrichment Analyses

GO enrichment analysis of the differentially expressed genes or lncRNA target genes was implemented by the GOseq R package, and gene length bias was corrected [45]. GO terms with corrected *p* values less than 0.05 were considered significantly enriched differentially expressed genes. We used KOBAS software [46] to test the statistical enrichment of the differentially expressed genes or lncRNA target genes in the KEGG pathways.

### 5.5. Target Gene Prediction

In the cis mode, lncRNAs act on neighboring target genes. We searched the coding genes that were 10 k/100 k upstream and downstream of the lncRNAs and then analyzed their function. Regarding their trans role, lncRNAs must be identified by their expression level compared with that of other relative genes. Although there were no more than 25 samples, we used custom scripts to calculate the expression correlations between the lncRNAs and coding genes.

### 5.6. Gene Expression Validation by qRT-PCR

The first strand cDNA was obtained by using a PrimeScript^TM^ First Strand cDNA Synthesis Kit (TaKaRa) with RNA (1 µg/mL). We selected five differentially expressed lncRNAs and their predicted target genes that may contribute to the transdifferentiation of cBMSCs (Appendix A). All primers used are shown in Appendix A. The *GAPDH* gene was used as an internal control. qRT-PCR was performed using a CFX Connect^TM^ Real Time PCR Detection System. The reaction mixture was prepared with Maxima SYBR Green/ROX qPCR Master Mix (2×) (Thermo Scientific, USA). The relative expression levels were calculated by the 2^−∆∆*C*t^ method.

## 6. Conclusions

To the best of our knowledge, this study is the first to focus on lncRNAs during the transdifferentiation of canine BMSCs, and lncRNAs expression changes were acquired and verified. In addition, an enrichment analysis of the targets of differentially expressed lncRNAs was performed, which could serve as a foundation for further mechanistic studies.

## Figures and Tables

**Figure 1 ijms-21-05549-f001:**
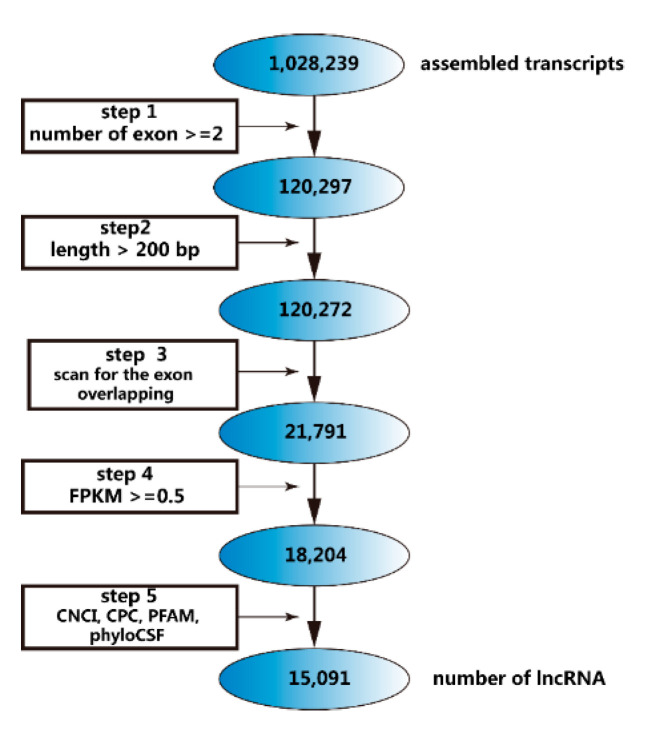
The five-step filter we used to screen long noncoding RNAs (lncRNAs) and the number of residual transcripts after each step was shown.

**Figure 2 ijms-21-05549-f002:**
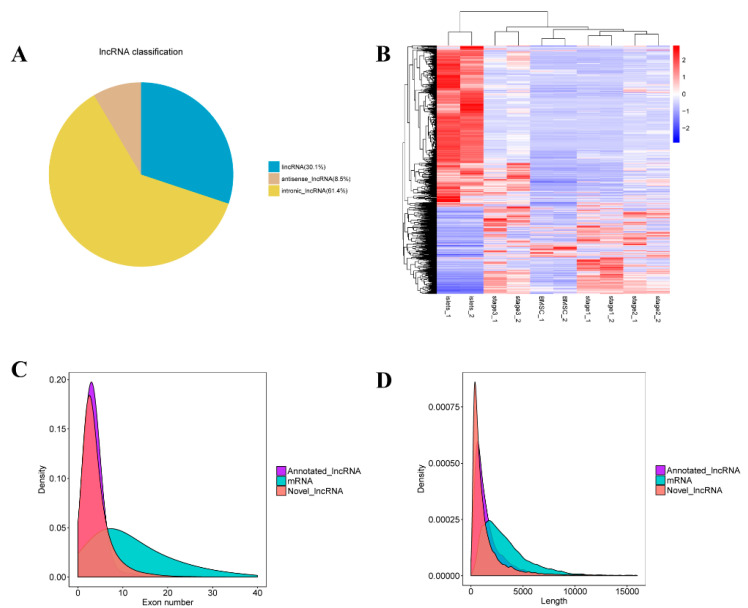
Overview of the lncRNA sequencing: (**A**) the percentage of different types of lncRNAs; (**B**) the hierarchical cluster diagram of differential transcript expression; (**C**) the exon number distribution of mRNA, annotated lncRNA, and novel lncRNA; and (**D**) the length distribution of mRNA, annotated lncRNA, and novel lncRNA.

**Figure 3 ijms-21-05549-f003:**
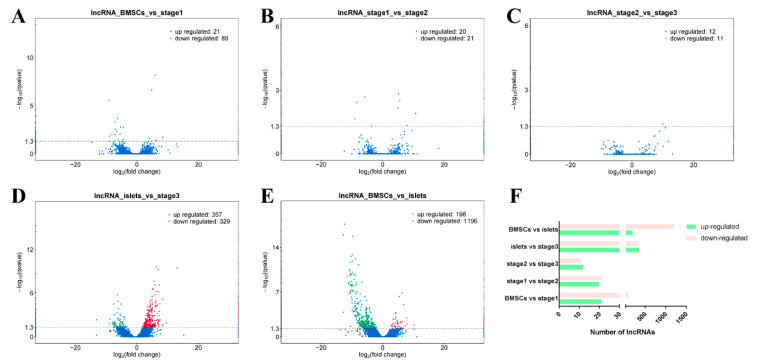
The volcano plot of differentially expressed lncRNAs of all comparison pairs: (**A**) For B vs. S1, there were 21 upregulated and 89 downregulated lncRNAs. (**B**) For S1 vs. S2, there were 20 upregulated and 21 downregulated lncRNAs. (**C**) For S2 vs. S3, there were 12 upregulated and 11 downregulated lncRNAs. (**D**) For I vs. S3, there were 357 upregulated and 329 downregulated lncRNAs. (**E**) For B vs. I, there were 198 upregulated and 1196 downregulated lncRNAs. (**F**) The statistics of all differentially expressed lncRNAs and the numbers were narrowed after induction.

**Figure 4 ijms-21-05549-f004:**
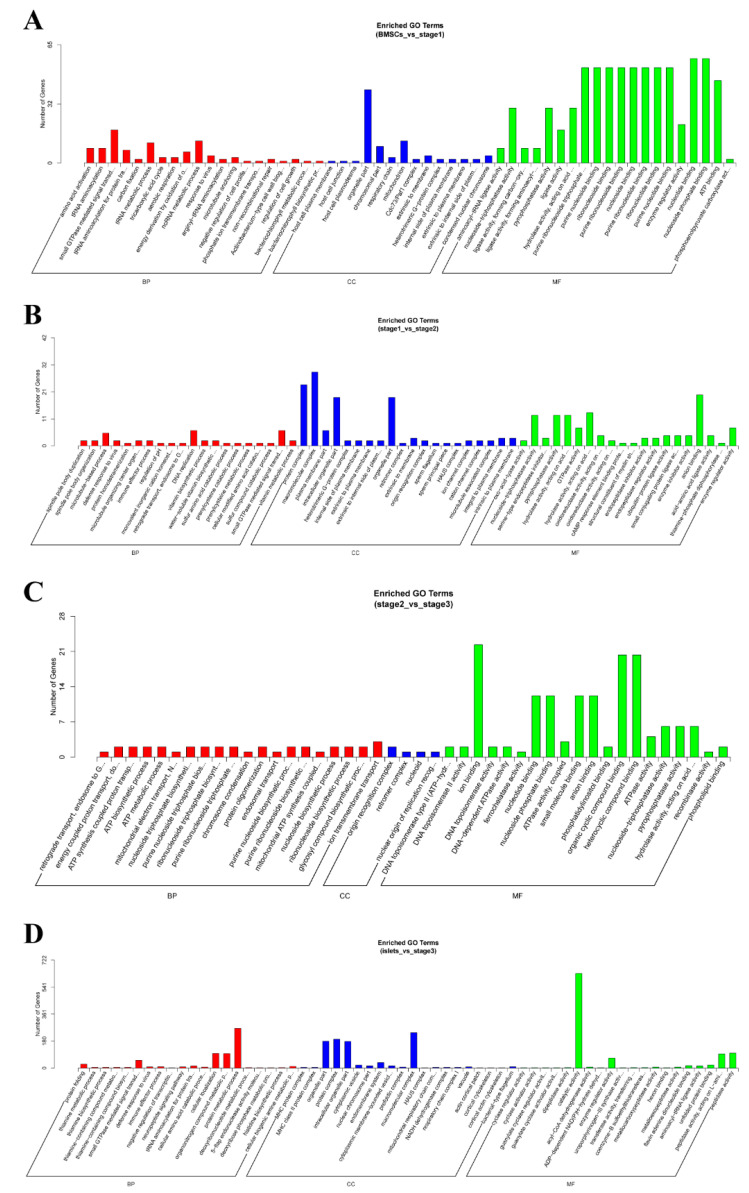
The Gene Ontology (GO) analysis of colocalized genes of differentially expressed lncRNAs. (**A**) The comparison of B vs. S1. (**B**) The comparison of S1 vs. S2. (**C**) The comparison of S2 vs. S3. (**D**) The comparison of I vs. S3.

**Figure 5 ijms-21-05549-f005:**
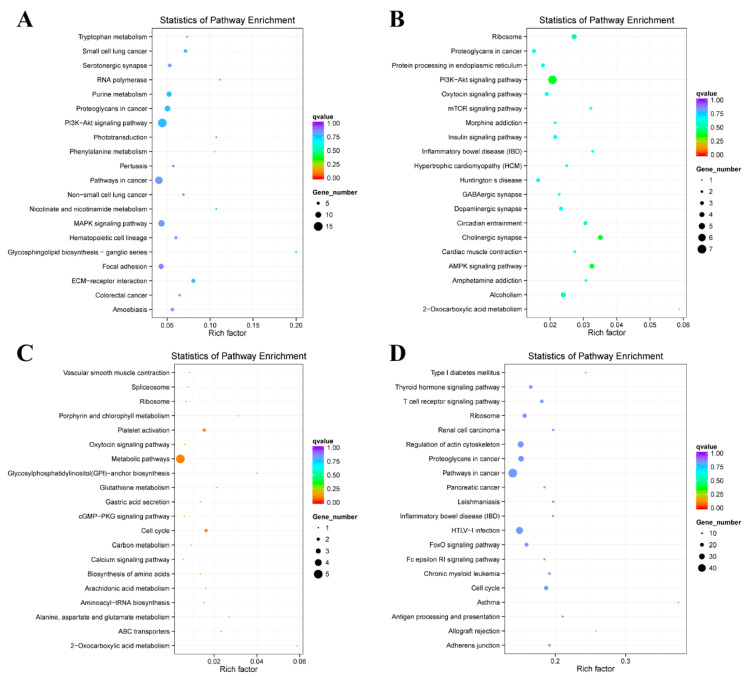
The Kyoto Encyclopedia of Genes and Genomes (KEGG) analysis of colocalized genes of differentially expressed lncRNAs. (**A**) The comparison of B vs. S1. (**B**) The comparison of S1 vs. S2. (**C**) The comparison of S2 vs. S3. (**D**) The comparison of I vs. S3.

**Figure 6 ijms-21-05549-f006:**
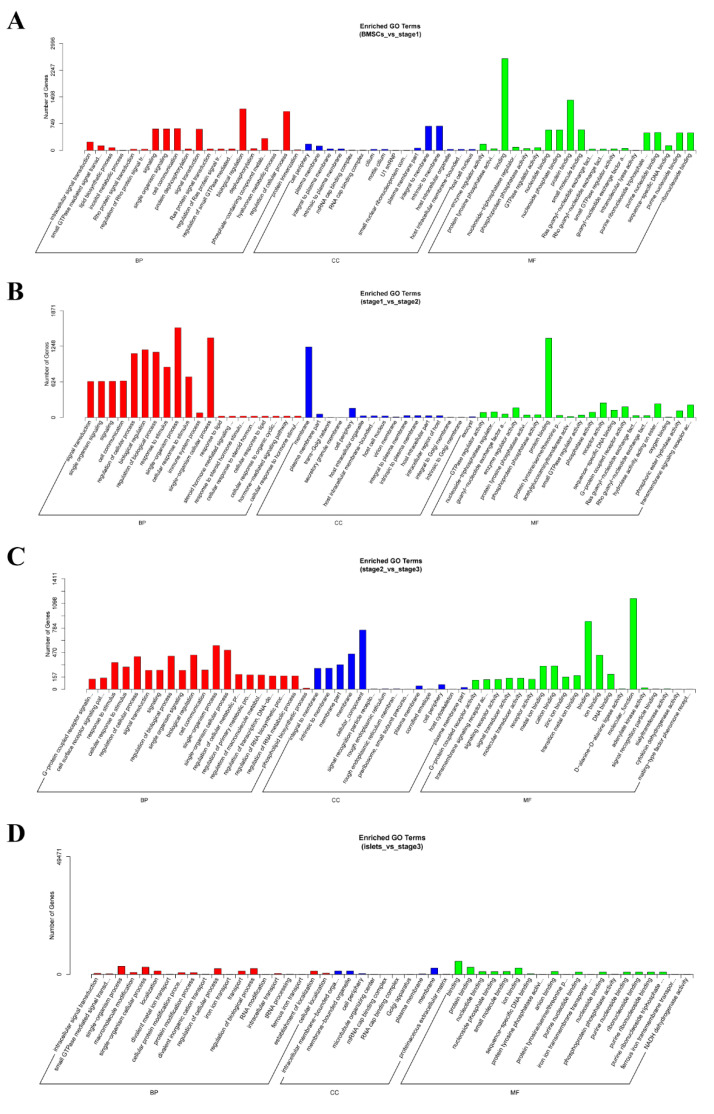
The GO analysis of co-expressed genes of differentially expressed lncRNAs. (**A**) The comparison of B vs. S1. (**B**) The comparison of S1 vs. S2. (**C**) The comparison of S2 vs. S3. (**D**) The comparison of I vs. S3.

**Figure 7 ijms-21-05549-f007:**
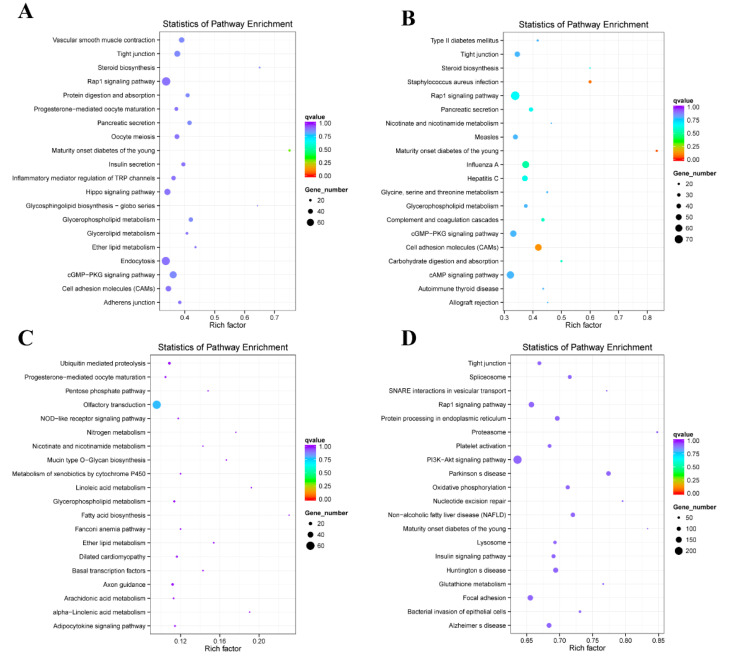
The KEGG analysis of co-expressed genes of differentially expressed lncRNAs. (**A**) The comparison of B vs. S1. (**B**) The comparison of S1 vs. S2. (**C**) The comparison of S2 vs. S3. (**D**) The comparison of I vs. S3.

**Figure 8 ijms-21-05549-f008:**
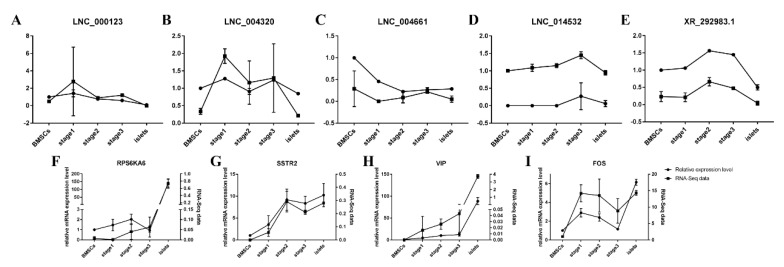
Validation of five differentially expressed lncRNAs (**A**–**E**) and four relative genes (**F**–**I**).

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
