# Peer review of "Genome-Wide Analysis Reveals Changes in Long Noncoding RNAs in the Differentiation of Canine BMSCs into Insulin-Producing Cells"

_ijms, 2020, doi:10.3390/ijms21155549_

Round 1

Reviewer 1 Report

In this study, authors hypothesize that lncRNAs are involved in bone marrow MSCs (BMSC) transdifferentiation to insulin producing cells (IPCs). For this authors utilized differentiation protocol with BMSCs - stage1- stage 2- stage3, IPCs to analyze differential lncRNAs expression. Of the 15,091 annotated lncRNAs on the microarray authors found differentially expressed genes. Authors further performed validation of targets by qPCR. It is interesting but generally a descriptive/ in silico study needs extensive solid wet-lab experimentation.

Major concerns include 

  1. Authors did not characterize MSCs or differentiated IPCs. This is important and needs to be performed to strengthen the current findings.
  2. Validation experiments should be plotted separately to understand what happens to a given lncRNA with time in the differentiation process.
  3. There is no stats applied on the qPCR database
  4. In vitro functional role of the identified lncRNA would be important piece of data to bring out lncRNAs role in IPCs.

Reviewer 2 Report

The manuscript written by Jinglu et al on the differentiation of canine BMSCs into insulin-producing cells and how genome-wide analysis revealed changes in long noncoding (lnc) RNAs is an interesting read. This study revealed how the lncRNA can play important role in important biological system, including differentiation of canine BMSCs. However, the article seems like a review article. Additional in vitro or in vivo experiments needs to be performed to validate how lncRNA regulate and being regulated during differentiation of canine BMSCs into insulin-producing cells. These experiments need to go beyond mere ‘GO’ and ‘KEGG’ pathway analysis.  

Few suggestions to improve the presentation of the manuscript is as follows.

  1. Line 39-41 needs to be re-written for clarity: “LncRNAs can be classified into two categories 39 based on whether they function in cis-, the lncRNAs of which are located around the 40 transcription site, or in trans-, the role of which is more complicated than the cis-role.”
  2. Line 43-44: “It has been reported that lncRNAs can be therapeutic targets of many diseases, such as breast 43 cancer [12] and hepatocellular carcinoma [13].” May be written as “It has been reported that lncRNAs can be therapeutic targets for managing many diseases, such as breast 43 cancer [12] and hepatocellular carcinoma [13].”
  3. The font size of figure legends in figure 2, 3, 5, 6, 7 and 8 should be bigger because it is difficult to read few of the legends.
  4. Figure 4 may be stretched so that there is enough gap between the legends. The font size of figure can be increased.
  5. Line 163: ‘constitute’ may be replaced by ‘is’.
  6. Line 192 and 193: “This enhanced the background of the canine lncRNA database…” may be written as “This provided additional information about canine lncRNA database…”

Line number 298: “…analysis .and writing…” may be written as “…analysis, and writing….”

Round 2

Reviewer 2 Report

Authors rectified previous raised concerns.